# APOBEC3-Mediated RNA Editing in Breast Cancer is Associated with Heightened Immune Activity and Improved Survival

**DOI:** 10.3390/ijms20225621

**Published:** 2019-11-10

**Authors:** Mariko Asaoka, Takashi Ishikawa, Kazuaki Takabe, Santosh K. Patnaik

**Affiliations:** 1Department of Breast Surgery, Roswell Park Comprehensive Cancer Center, Buffalo, NY 14263, USA; 2Department of Breast Surgery and Oncology, Tokyo Medical University, Tokyo 160-8402, Japan; 3Department of Surgery, Jacobs School of Medicine and Biomedical Sciences, State University of New York, Buffalo, NY 14263, USA; 4Department of Surgery, Niigata University Graduate School of Medical and Dental Sciences, Niigata 951-8510, Japan; 5Department of Surgery, Yokohama City University, Yokohama 236-0004, Japan; 6Department of Thoracic Surgery, Roswell Park Comprehensive Cancer Center, Buffalo, NY 14263, USA

**Keywords:** APOBEC, breast cancer, RNA editing, sequencing, tumor immune microenvironment

## Abstract

APOBEC3 enzymes contribute significantly to DNA mutagenesis in cancer. These enzymes are also capable of converting C bases at specific positions of RNAs to U. However, the prevalence and significance of this C-to-U RNA editing in any cancer is currently unknown. We developed a bioinformatics workflow to determine RNA editing levels at known APOBEC3-mediated RNA editing sites using exome and mRNA sequencing data of 1040 breast cancer tumors. Although reliable editing determinations were limited due to sequencing depth, editing was observed in both tumor and adjacent normal tissues. For 440 sites (411 genes), editing was determinable for ≥5 tumors, with editing occurring in 0.6%–100% of tumors (mean 20%, SD 14%) at an average level of 0.6%–20% (mean 7%, SD 4%). Compared to tumors with low RNA editing, editing-high tumors had enriched expression of immune-related gene sets, and higher T cell and M1 macrophage infiltration, B and T cell receptor diversity, and immune cytolytic activity. Concordant with this, patients with increased RNA editing in tumors had better disease- and progression-free survivals (hazard ratio = 1.67–1.75, *p* < 0.05). Our study identifies that APOBEC3-mediated RNA editing occurs in breast cancer tumors and is positively associated with elevated immune activity and improved survival.

## 1. Introduction

Base modifications by deamination of adenine to inosine (A-to-I) and of cytidine to uracil (C-to-U) are the major types of RNA editing in higher eukaryotes. Inosine and uracil in RNA are read as guanosine and thymine, respectively, during RNA translation. Thus, RNA editing has the capability of altering protein sequences encoded by DNA. The activation-induced deaminase (AID), APOBEC, and cytidine deaminase (CDA) proteins of mammals harbor the cytidine deaminase motif for hydrolytic deamination of C to U [1]. CDA is involved in the pyrimidine salvaging pathway. While AID causes C-to-U mutation of DNA, multiple studies have failed to identify any RNA editing activity of the protein [2]. Humans have ten (*APOBEC1, 2*, *3A–D*, *3F–H*, and *4*) and mice have four (*APOBEC1–4*) *APOBEC* genes. Expression of *APOBEC1*, *APOBEC2*, and *APOBEC4* genes is largely restricted to intestine, muscle, and gonad tissues, respectively, whereas *APOBEC3* genes are expressed the most in cells of the immune system [3]. RNA sequencing studies have identified wide-spread C-to-U RNA editing in mammals (e.g., [4,5,6]).

APOBEC3 proteins cause C-to-U deamination of single-stranded DNA, and this DNA deamination activity underlies the proteins′ well-established role in restricting viruses and retrotransposon elements [7,8]. Lately, the DNA deamination activity of APOBEC3 proteins has also been identified as a major contributor to mutagenesis in multiple human cancers, including that of breast [9,10,11,12]. Furthermore, individuals with particular structural changes in *APOBEC3A* and *APOBEC3B* genes have been found to be at an increased risk for development of breast, ovarian, and liver cancer [13,14].

While all human APOBEC3 proteins have been known for a while to bind RNA [15], it is only recently that their C-to-U RNA editing activity has come to light. In both independent and collaborative studies, our group has demonstrated such hitherto unknown C-to-U RNA editing activities for human APOBEC3A, APOBEC3B, and APOBEC3G enzymes, and shown physiologically-affected occurrence of such APOBEC3-mediated RNA editing in immune cells [16,17,18,19]. Although the prevalence and significance of APOBEC3-mediated DNA mutagenesis in multiple types of cancer are now known, such knowledge is lacking for APOBEC3-mediated RNA editing for any cancer. Because APOBEC3s are well-studied for breast cancer (e.g., [12,20,21,22,23]), we sought to examine this cancer to survey the prevalence of APOBEC3-mediated RNA editing and investigate its biological and clinical relevancies.

## 2. Results

### 2.1. Compilation of a Set of APOBEC3-Mediated RNA Editing Sites

In order to estimate APOBEC3-mediated C-to-U RNA editing in tissues of The Cancer Genome Atlas (TCGA)-breast carcinoma (BRCA) cohort, we collected a list of genome positions (editing sites) for which C-to-U RNA editing by APOBEC3 enzymes is known to occur in human cells. Monocytes and macrophages express multiple *APOBEC3* but not *APOBEC1*, *APOBEC2*, or *APOBEC4* genes [3]. Except for 96 sites identified by us in HEK-293T/APOBEC3B transfectants (manuscript in preparation), the sites were collected from published data. In all these studies, RNA editing at the sites was established using RNA sequencing-based approaches, although allele-specific polymerase chain reaction (PCR) or Sanger sequencing of cDNA amplicons was also utilized for a few dozen sites.

A total of 5208 sites, which were all the APOBEC3-mediated editing sites that were known when we performed this investigation, were thus collected (Appendix A). A majority of the sites (79%) were identified with HEK-293T/APOBEC3A transfectants (Figure 1a). A small fraction (5.5%) had been discovered in multiple studies. Most (93.9%) of the sites lie within coding (62.7%) or untranslated (31.2%) regions of mRNAs. C-to-U editing of the mRNAs is predicted to affect their encoded protein sequences for a significant number of sites (29.7%), through either non-synonymous codon change (1195 sites) or gain of a stop codon (353 sites; Figure 1a).

### 2.2. Estimation of RNA Editing Levels in TCGA-BRCA Tissues

We identified 1040 breast cancer tumor and 93 adjacent normal tissues in the TCGA-BRCA project for which RNA and DNA (whole exome) sequencings had been performed with the same tissue portion. TCGA sequencing data that had been aligned with the reference human genome was obtained from Genomic Data Commons (GDC) portal and analyzed to estimate C-to-U RNA editing levels at each of the 5208 sites. A bioinformatics pipeline was developed for this study, the schematics of which is shown in Appendix A.

The exact criteria used for the RNA editing determination are noted in the Materials and Methods section. These criteria were developed such that the degree of their stringency permitted reliably confident RNA editing determination considering the type of data that was available to us. Nevertheless, RNA editing could not be calculated for a large fraction (97%) of determinations due to inadequate sequencing depth in the data. This can be seen in the information provided in Appendix A, which lists criteria and their fulfillment in the editing determination workflow.

C-to-U editing determinations could not be made for any sample for 664 of the 5208 known APOBEC3-mediated editing sites. For the remaining 4544 sites, editing was determinable for 1–371 samples (mean = 11, standard deviation (SD) = 7). Among tumors, editing determination was possible for ≥5 tumors for only 2082 sites. A total of 1625 C-to-U editing events (editing level >0) were identified with the workflow. With a set of 198 negative control sites that are transcribed as C but are not known to undergo APOBEC3-mediated C-to-U RNA editing (Appendix A), 21 editing events were observed. To further affirm the strength of the editing determination workflow, the 5208 editing sites were also evaluated for the hypothetical C-to-A type of RNA editing and only 152 such events were identifiable.

Editing determinations that were programmatically made with the bioinformatics pipeline were confirmed for multiple determinations through visual assessment of aligned RNA and DNA sequencing data. A few examples of such visualization of editing events are depicted in Appendix A. Because of the unavailability of TCGA tissues or their nucleic acid preparations, validation of editing through experimental methods such as allele-specific PCR or Sanger sequencing could not be performed. To account for the possibility of misestimation of editing due to artifacts arising from read sequence alignment, we reassessed C-to-U RNA editing levels for 24 randomly selected editing events after aligning RNA sequencing data with a different mapping software (Subread subjunc [24], instead of STAR, which was used for the GDC data). As shown in Appendix A, editing levels with the originally aligned and Subread-aligned sequencing data had good correlation (Pearson *r* = 0.85), with a mean deviation value of 13.6%.

### 2.3. APOBEC3-Mediated C-to-U RNA Editing is Prevalent in Breast Cancer Tumors

C-to-U RNA editing events at the 5208 known APOBEC3-mediated editing sites were identified among the TCGA-BRCA tissues. For 440 sites, in a total of 411 gene transcripts, editing was determinable for ≥5 tumors, with editing occurring at the sites in 0.6%–100% of the tumors (mean = 20%, SD = 14%) at an average level of 0.6%–20% (mean = 7%, SD = 4%). Distributions of these values are plotted in Figure 1b. Among these 440 sites, editing was most prevalent for *GATAD2B* (100% of tumors with a determinable editing level), *SERPINA1* (81%), and *AMPD3* (80%) RNAs. Mean editing levels were highest for *SH3PXD2A* (20%), *FAM208A* (20%), and *DSCR3* (19%) RNAs. The number of tumors with RNA editing was largest for *SERPINA1* (160), followed by *DDOST* (60) and *EVI2B* (43). Mean editing levels and editing frequency among tumors for the 30 RNAs for which editing was most prevalent among tumors are plotted in Figure 1c. Appendix A has genome coordinates and other annotations for the 50 most commonly edited sites among tumors. Of the 411 genes representing the 440 sites, 80 are in the gene ontology (GO) biological process set of genes for “immune system process,” which is a 1.5-fold overrepresentation with false discovery rate (FDR) of 0.03.

No editing event was observed for 409 tumors and 32 normal tissues that had determinable editing for ≥5 sites. Among the 448 tumors and 44 normal tissues that had editing at ≥1 site and ≥5 editing-determinable sites, an average of 6.6% of determinable sites had editing (SD = 4.2%), with an average editing level of 6.6% (SD = 3.4%). To summarize editing levels across multiple editing sites in a sample, we used the ratio of editing-positive sites and determinable sites as an editing score. This editing score was calculated only for samples with ≥5 editing-determinable sites. Samples with editing scores of 0 and >0 were considered as editing-low and editing-high samples, respectively, for further analyses. There were thus 409 editing-low and 448 editing-high tumors in the TCGA-BRCA cohort.

Tumor and normal tissues had similar editing scores as suggested by *p*-value of 0.06 in paired analysis with standard *t*-test. The incidence of editing-high tumors was similar among groups of tumors of the American Joint Committee on Cancer pathological stages I–IV (Figure 2). Incidence rates were also similar between lymph node metastasis-negative (TNM N0) and -positive (TNM N1–3) tumors. Incidence rates were significantly different among TNM T1–T4 tumor groups, and among different breast cancer subtypes, with editing-high tumors about 40% more prevalent for HER2 compared to luminal subtype. Both estrogen and progesterone receptor negativity were associated with increased editing-high incidence, whereas the fraction of editing-high tumors was less among HER2 receptor-negative compared to -positive tumors. High mitosis and nuclear pleomorphism scores were associated with increased editing-high incidence. The relation of editing with cell proliferation that was suggested by this finding was also noted with the observation of increased editing-high incidence among tumors with high gene expression of *MKI67*, a well-known proliferation marker (Figure 2).

### 2.4. APOBEC3 Gene Expression Correlates with RNA Editing Better Than APOBEC-Mediated Mutations

As expected, editing scores were associated with tumor gene expression of *APOBEC3* genes. Expressions among editing-high tumors were significantly raised for all seven *APOBEC3*s, with the largest expression difference (about 3×) seen for *APOBEC3A* (Figure 3a). In line with this, editing scores had the best correlation with *APOBEC3A* gene expression (Pearson *r* = 0.31; Figure 3b). Association of *APOBEC3* gene expression with editing at specific sites was also examined. In general, tumors with editing at specific sites had higher expression of multiple *APOBEC3* genes compared to tumors without editing. Representative examples are shown in Appendix A. While positive correlations of *APOBEC3* gene expressions and RNA editing were modest, except for *APOBEC3B* (*r* < 0), they were decidedly higher than those observed for APOBEC-mediated DNA mutation burden of tumors (Figure 3b). Consistent with this, editing-high and -low tumors had similar single nucleotide mutation burden, APOBEC-mediated as well as all, and predicted neoantigen load (Figure 3c). However, editing-high tumors had more genome copy number alterations, aneuploidy, intra-tumoral clonal heterogeneity, and homologous recombination deficiency than editing-low tumors.

### 2.5. RNA Editing is Associated with Enriched Expression of Immune- but not Cancer Development-Related Gene Sets

To obtain an insight into the biological basis for the variability in RNA editing among the TCGA-BRCA tumors, a differential gene expression analysis was performed to compare editing-high and -low tumor groups. Expression of 66 and 20 genes was significantly up- and downregulated with a ≥2-fold change, respectively, in the editing-high compared to -low group (adjusted *p* < 0.05 in limma′s Bayes-moderated *t*-test). *APOBEC3A* was the most upregulated gene (Appendix A); other upregulated genes included those encoding chemokine ligands, such as *CXCL10* and *CXCL13*, and interferons, such as *IFNG* and *IFNL1*. Of the 66 upregulated genes, 44 are in the PANTHER GO) biological process dataset of genes for immune system process, a 5.1-fold overrepresentation with FDR < 2e-20. Consistent with this, gene set enrichment analysis using the gene set variation analysis (GSVA) method [25] showed that, compared to the editing-low group, the editing-high group had significantly enriched expression for six of the seven immune-related gene sets of the Molecular Signatures Database (MSigDB) Hallmark collection (FDR < 0.05; Figure 4a). In contrast, enrichment was not observed for gene sets that are typically associated with cancer development, such as those for angiogenesis and epithelial mesenchymal transition (Figure 4b). Overall, enrichment was observed for 22 of the 50 hallmark gene sets.

### 2.6. Immune Activity is Elevated in Tumors with High RNA Editing

The immune response-related results of gene expression analyses that are described above, and the overrepresentation of immune response genes among the genes undergoing RNA editing in tumors, suggested a relevance of RNA editing in the tumor microenvironment. To test this, we compared editing-high and -low tumor groups for various features of the immune microenvironment. Tumor-infiltrating immune cell subsets were quantified from tumor gene expression data using the CIBERSORT algorithm [26]. Although both editing-high and -low tumor groups were similar for overall lymphocytic infiltration, the editing-high tumors had significantly more T cells as well as various T cell subtypes (CD8^+^, CD4^+^, T-reg, and gamma delta T; fold changes of 1.08–1.67). On the other hand, the fraction of B cells was reduced compared to editing-low tumors (fold change of 0.88; Figure 5a). Similarly, while the two groups were similar for macrophages, the M1 subtype of macrophages was more prevalent in editing-high group (fold change of 1.21), whereas the M2 subtype was less prevalent (fold change of 0.88) compared to the editing-low group (Figure 5a).

As may be expected from the positive association of RNA editing with T cell and M1 macrophage infiltration, RNA editing-high tumors showed markers of heightened anticancer immune response. Both B cell receptor (BCR) and T cell receptor (TCR) diversity (Shannon scores) and immune cytolytic activity (CYT score) were significantly higher (fold changes of 1.14–1.89) in the editing-high group of tumors (Figure 5b).

### 2.7. Patients with Tumors with High RNA Editing in Tumors have Better Survival

The finding of a heightened immune activity in editing-high tumors indicated that patients who have high RNA editing in tumors may have a better clinical outcome. To investigate this, we performed survival analyses to compare the editing-low and -high groups of tumors. Recently released, uniformly curated, and filtered TCGA data for survival endpoints was used for this analysis [27]. Both disease-free and progression-free survivals were better for patients with editing-high tumors compared to those with editing-low tumors (Figure 6). Log-rank *p*-values in this analysis were <0.05, and hazard ratio (HR) values in Cox regression models were 1.75 and 1.67, respectively. For disease-specific and overall survival as well, the trend of improved survival for the editing-high group was observed, although *p*-values were >0.05 (both 0.08).

## 3. Discussion

Base conversions in RNAs through editing by enzymes like APOBEC3s can have a consequence at the protein level because of an effect of editing on not only codon sequences but also on intron splicing, RNA stability, microRNA targeting, etc. Resulting changes in phenotypes of cancerous cells as well as cells of the tumor microenvironment therefore have the potential to affect cancer biology (e.g., [28,29]). These changes may include the generation of neoantigens on cancer cells and alterations in anticancer response pathways in immune cells [30]. The A-to-I type of RNA editing has been examined in tumors of multiple cancers, including breast cancer [31,32,33]. For C-to-U RNA editing, our knowledge of its prevalence in cancers is limited. APOBEC1-mediated RNA editing has been studied for one cancer [34], and cancers have not been surveyed for the prevalence of APOBEC3-mediated C-to-U RNA editing. We therefore investigated the prevalence and significance of such editing in tumors of breast cancer, for which APOBEC3-mediated DNA mutagenesis has been substantially explored. APOBEC3s are significant contributors of DNA mutagenesis in cancers, including breast cancer [35]. Human APOBEC3A, APOBEC3B, and APOBEC3G possess RNA editing capability [16,17,18,19], whereas it is unknown if the other APOBEC3 proteins too have such ability.

Our study, the first examination of APOBEC3-mediated C-to-U RNA editing in cancer, identifies that this phenomenon occurs in breast cancer tumors. Transcriptome and exome sequencing data were examined for this. Stringent criteria were used in the study to identify such editing in order to reduce false positivity because of sequence variations arising from inherited DNA polymorphisms, DNA mutations, sequencing errors, and read sequence alignment artifacts. Strictness of the criteria was limited by the depth of coverage in the aligned sequencing data that was available to us. With the criteria that were used, approximately 97% of editing estimations were indeterminable because of shallow coverage (Appendix A). Considering only base calls of high quality (phred33 ≥20), the median coverage at the examined genome positions was 72 and 31, respectively, for RNA and DNA sequencing data (mean = 195 and 56, respectively). Nevertheless, C-to-U RNA editing events could be identified in the tumor tissues. Editing was observed for 448 tumors with editing determinable at ≥5 sites, and editing occurring at average 20% of the determinable sites at an average level of 7%. Reliability of the editing estimations is also likely to be high because the identified C-to-U editing events were at sites known to undergo APOBEC3-mediated RNA editing. Reliability was also attested through assessments for the hypothetical C-to-A type of RNA editing at the known C-to-U editing sites, and of C-to-U editing at a negative control set of genome positions that are not known to undergo APOBEC3-mediated RNA editing. While editing determinations could be somewhat validated through the examination of RNA sequencing data after realigning it with a different alignment software (Appendix A), validation with a different experimental technique was not possible because of unavailability of specimens. We were also unable to find an independent breast cancer tumor cohort of sufficient size with both RNA and exome or genome sequencing data of adequate depth coverage for validation purposes.

We devised an editing score to simplify further examination of the multi-dimensional RNA editing determinations. RNA editing across multiple sites of a sample was summarized into the editing score for the sample. The score was simply the fraction of sites for which editing was observed in the sample. We chose not to use the mean of editing levels among the sample′s sites because the precision of editing level calculations was not high due to limitations of depth of sequencing coverage. The approach of targeted sequencing with a focus on known editing sites, which permits much deeper coverage, should be considered in future studies for accurate determination of RNA editing levels.

As expected, tumors with high editing scores (value >0) had higher expression of multiple *APOBEC3* genes compared to tumors with a score of 0 (Figure 3a). This association of editing with higher *APOBEC3* gene expression was also seen in the examination of editing at specific sites (Appendix A). Interestingly, the correlations of tumor *APOBEC3* gene expression with burden of mutations with APOBEC signature [12] were poorer compared to the RNA editing score (Figure 3b). This may be expected because, unlike DNA mutation (which gets fixed in the genome), edited RNAs have a limited life and their detection is contingent upon concurrent expression of RNA editing enzymes.

Because of the constrained site-specific editing determinability, determination being possible for an average of only 11 sites among tumors, we did not examine the biological or clinical relevance of editing of individual RNAs. In examinations using the editing score as an overall marker of APOBEC3-mediated RNA editing, numerous insights were obtained. Tumors with increased RNA editing had more cancer-related changes such as aneuploidy and copy number aberrations in their genomes (Figure 3c). Whether the increase in editing is in response to such changes remains unknown.

Not only were immune-related genes overrepresented among genes for which editing was detected, but differential gene expression and gene set enrichment analyses also revealed that expression of numerous genes associated with immune response was higher among tumors with high scores in contrast to tumors with low scores. Furthermore, tumors with high editing had more T compared to B cells. Infiltration levels of multiple subtypes of T cells, including CD4^+^ and CD8^+^, were higher among these tumors (Figure 5a). Not only was T cell infiltration high, their activity was also likely high as indicated by the increased T cell receptor diversity and cytolytic activity that were observed in the editing-high tumors (Figure 5b). These tumors also had more M1 and less M2 macrophages. This is consistent with the known observation that both *APOBEC3* gene expression and C-to-U RNA editing are markedly elevated in M1 compared to M2 macrophages from peripheral blood [18]. Concordant with these indications of a heightened anticancer immune response in the tumor microenvironment, patient survival was significantly better for those whose tumors had high RNA editing (Figure 6).

In this study, we used a well-established method, CIBERSORT, to estimate tumor-infiltrating immune cells using RNA sequencing data of whole tumor, which, besides cancer cells, contains unknown cells as well as closely related types of cells such as immune cell subsets. Accurate estimation of tumor immune cells with this algorithm has limitations because of inadequacies in cell-type signatures that the algorithm relies on, phenotypic malleability, and disease-induced alterations in cellular differentiation or transcription. Therefore, abundances of some specific tumor-infiltrating immune cell types could have been over- or under-estimated in our study, and the associations of abundances with RNA editing that we observed may be different if infiltrating immune cells had been quantified by flow cytometric or histologic methods.

The findings of our study imply that APOBEC3 enzymes are relevant in breast cancer because of not only their DNA mutagenic but also their RNA editing activity, and they highlight the pertinence of finer dissection of such editing in further studies. While we could detect C-to-U RNA editing events in the tumors, the cellular origin of such events remains unclear. Overrepresentation of edited sites among immune-related genes, enriched expression of such genes in editing-high tumors, and the fact that *APOBEC3* gene expression is much higher among immune compared to epithelial cells [7] suggests that the editing occurs in the tumor microenvironment. Further investigations implementing methods such as single-cell sequencing and isolation of sub-populations of cells from tumors are needed to definitively know if the editing occurs in cancerous epithelial or immune cells of breast tumors. The biological consequences of the editing events on cancer development, progression, and immune response also remain unknown.

## 4. Materials and Methods

### 4.1. Collation of APOBEC3-Mediated C-to-U RNA Editing Sites

A total of 5208 sites of 3630 gene transcripts known to undergo APOBEC3-mediated C-to-U RNA editing were identified from our published and ongoing studies [16,17,18,19]. Specifically, genomic coordinates for the sites were collated from Appendix A of the publications. C-to-U RNA editing at these sites was observed in human 293T fibroblasts that were transiently transfected with expression constructs for human APOBEC3A, APOBEC3B, or APOBEC3G, and/or in human monocytes and macrophages that, respectively, were subjected to hypoxia and M1 polarization. The LiftOver tool at https://genome.ucsc.edu/cgi-bin/hgLiftOver was used to convert the published GRCh37 human genome reference-based coordinates to GRCh38 in the case of sites identified in macrophages or monocytes [18]. The list of 5208 sites is provided in Appendix A. The ANNOVAR tool (Feb. 2016 version) and its RefSeq-based database (20151211 build) were used to annotate the sites with information on gene name, RNA region, effect of editing on protein sequence, etc. For negative control examination, 198 genome positions that are transcribed as C and are adjacent to one of the 5208 editing sites but are not known to undergo APOBEC3-mediated RNA editing were randomly selected. These control sites are listed in Appendix A.

### 4.2. Access to Controlled Sequence Alignment Data of TCGA Project

Approval for access to aligned DNA (whole exome) and RNA sequencing data was obtained from National Institutes of Health, Bethesda, MD, USA, under study accession phs000178 of Database of Genotypes and Phenotypes (dbGaP). The data, in BAM format, was downloaded in early 2018 from the Genomic Data Commons (GDC) portal of National Cancer Institute, Bethesda, MD, USA, using the gdc-client GDC data transfer tool (version 1.3). Data in GDC were processed for all TCGA samples in a harmonized and uniform manner, using the GRCh38.p0 human genome reference [36]. Briefly, alignment of RNA sequencing data was done with STAR software (version 2.4) in a 2-pass workflow [37]. For whole exome DNA sequencing reads, BWA aligner software (version 0.7.12) was used in a workflow that used Picard MarkDuplicates (version 1.1.38) and GATK IndelRealigner (version 3.5) for duplicate-marking and co-cleaning [38].

### 4.3. Estimation of RNA Editing Levels in TCGA Tissues

Aligned DNA and RNA sequencing data were examined and compared to determine editing levels in 1040 tumor and 93 adjacent normal tissues. DNA and RNA data were from the same portion of TCGA tissue samples as determined from their biospecimen data identifiers. The data were analyzed with samtools (version 1.3) to generate pileups of base calls of phred33 quality ≥20 for each examined genome position. Pileups were then parsed with custom scripts in awk and R languages to count base calls (A/C/G/T) and to determine editing level from the count data, as outlined in Appendix A. In the examination for C-to-U RNA editing, reference and variant bases were C and T for genes that are transcribed from the + chromosomal strand, and G and A otherwise. First, RNA data were examined. Editing was deemed not determinable if coverage (sum of all base calls, *N*) was poor (<10), or if the reference base count (*R*) suggested the presence of DNA polymorphism, either homozygous (*R* < 6 for *N* ≤ 20, or < 0.4× *N* for *N* > 20) or heterozygous (*R* < 0.8× *N*, otherwise), or if variant base count (*V*) was 1. For *V* = 0, editing level was estimated as 0 if *R* was > 3× mean of *R*/*V* ratios of all samples with *V* > 0; otherwise, editing was considered indeterminable. For *V* ≥ 2, editing was indeterminable if DNA sequencing coverage was <10, or if the *p*-value in the comparison of *R* and *V* values of RNA and DNA data with one-sided Barnard′s test (Boschloo method) was >0.05. Otherwise, editing level was estimated as the *V*/(*R* + *V*) fraction if *V* was 0 for DNA data or *V*/(*R* + *V*) of RNA data was ≥10× that of DNA data, and, if *V* was > 2, there was no strong (>8×) or significant (Fisher exact test *p* ≤ 0.01) RNA sequencing strand bias of variant compared to reference bases. Strand bias was not tested for *V* = 2. Editing was considered indeterminable for all other cases. Appendix A lists these various criteria along with the number of RNA editing determinations made for the 1133 tissues and 5208 sites that fulfilled the criteria.

### 4.4. Validation of RNA Editing Level Estimates Using a Different Sequencing Read Alignment

For 24 C-to-U RNA editing events that were identified in four tumor samples, the RNA editing levels were redetermined after realigning RNA sequencing data. For realignment, samtools fastq utility was used to extract read data from the aligned data that had been generated by GDC. Only paired read sequences with concordant mapping were extracted. The sequences were then realigned using Subread [24] subjunc aligner software (version 1.5.0-p1) against version 81 of the Ensembl GRCh38 reference genome. With the realigned data, C-to-U RNA editing level was estimated as the ratio of variant to sum of variant and reference base calls of phred33 quality ≥20.

### 4.5. Collection of Clinical and Pathologic Information of TCGA Subjects

Tumor histologic scores for mitotic counts, nuclear pleomorphism, and tubular formation as per the Nottingham system were collected through manual examination of pathology reports with the aid of data and the search engine in the Text Information Extraction System (TIES) Cancer Research Network [39], a federated network of four cancer centers in the USA. All three scores could be recorded for 591 of the total 1093 TCGA-BRCA patients; at least one or at least two types of scores, respectively, could be recorded for 669 and 625 patients. TCGA Pan-Cancer Clinical Data Resource [27], the standardized and curated dataset for survival endpoints for the TCGA project, was the source for survival data for the TCGA-BRCA patients. Information for other clinical and pathologic variables was obtained from TCGA through the cBio Cancer Genomics Portal [40] in late 2017.

### 4.6. Quantification of TCGA Tumor Genome Features

Scores for intratumoral clonal heterogeneity, as well as tumor aneuploidy, copy number changes, homologous recombination deficiency, and neoantigen load were collated from a TCGA Pan-Cancer Atlas study [41]. To count all single nucleotide substitution mutations and obtain a minimum estimate of counts of mutations attributable to DNA deaminase activity of APOBEC enzymes as per the P-MACD method [12], publicly available mutation annotation format (MAF) files from TCGA were analyzed with the maftools [42] Bioconductor package (version 1.8.0).

### 4.7. Generation of TCGA Tumor Gene Expression Dataset

Gene-level counts of mapped RNA sequencing reads were obtained from the GDC portal with the TCGAbiolinks (version 2.5.9) Bioconductor package [43] in mid-2018. The counts were generated with HTSeq software (version 0.6.1-p1) with Gencode 22 annotation and Ensembl gene identifiers. To generate gene expression values in transcripts per million (TPM) units, counts were divided by total exon length of the identifiers. Corresponding Human Genome Organization (HUGO) gene symbols for the gene identifiers were identified with biomaRt (version 2.38.0) Bioconductor package [44] and Ensembl human genome annotation (version 92). In the case of multiple identifiers with the same symbol, TPM values were aggregated by addition. Gene identifiers without an identifiable gene symbol were removed. For log_2_-transformation, TPM values were padded with 0.05.

### 4.8. Analyses of Gene Expression Data

Log_2_-transformed TPM values were used. To identify genes that were differentially expressed between two groups, the empirical Bayes-moderated *t*-test in limma (version 3.38.2) Bioconductor package [45] was used with threshold values of 0.05 and 2, respectively, for *p*-value and absolute fold change value. To determine the enrichment of expression of the 50 gene sets of the Molecular Signatures Database (MSigDB) Hallmark collection (version 6.2), the gene set variation analysis (GSVA) method implemented in the GSVA Bioconductor package (version 1.30.0) was used [25]. Two-group comparison of GSVA scores were with limma′s *t*-test. To control false discovery in multi-testings of both differential gene expression and GSVA analyses, the Benjamini–Hochberg method was used to adjust raw *p*-values.

### 4.9. Measurements of TCGA Tumor Immune Microenvironment

Immune cytolytic activity in tumors was quantified from perforin 1 (*PRF1*) and granzyme A (*GZMA*) gene expression values as the CYT score [46]. Relative fractions of 22 types of tumor-infiltrating immune cells were inferred from tumor gene expression data (TPM values) with the CIBERSORT method. The online tool at https://cibersort.stanford.edu (jar version 1.06) was used with LM22 signatures and 100 permutations [26]. Values for certain types of cells were aggregated to generate relative fraction values for more comprehensive cellular classes such as lymphocytes [41]. The immune cell composition estimates for tumors for which the CIBERSORT *p*-value was ≥0.05 (about 10% of cases) were excluded from analysis. Other immune-related measurements of tumors such as B and T cell receptor diversities were from the study of Thorsson et al. [41].

### 4.10. Other

Unless noted otherwise, default option values were used for all software, group comparisons of continuous and categorical variables were with standard *t*-tests and Fisher′s exact tests, respectively, and *p*-values below 0.05 were considered significant. Analyses and visualization of data were performed using Prism (version 7.0d; GraphPad Software^®^, San Diego, CA, USA) and R (version 3.5+). The survival package (version 2.44) for R was used for survival analyses with Cox regression and log-rank tests for the determination of HR and *p*-values. Statistical overrepresentation tests of gene lists with Fisher′s exact test and false discovery calculation were conducted online at https://pantherdb.org with the complete GO biological process annotation dataset of PANTHER [47] database (version 14.1).

## Figures and Tables

**Figure 1 ijms-20-05621-f001:**
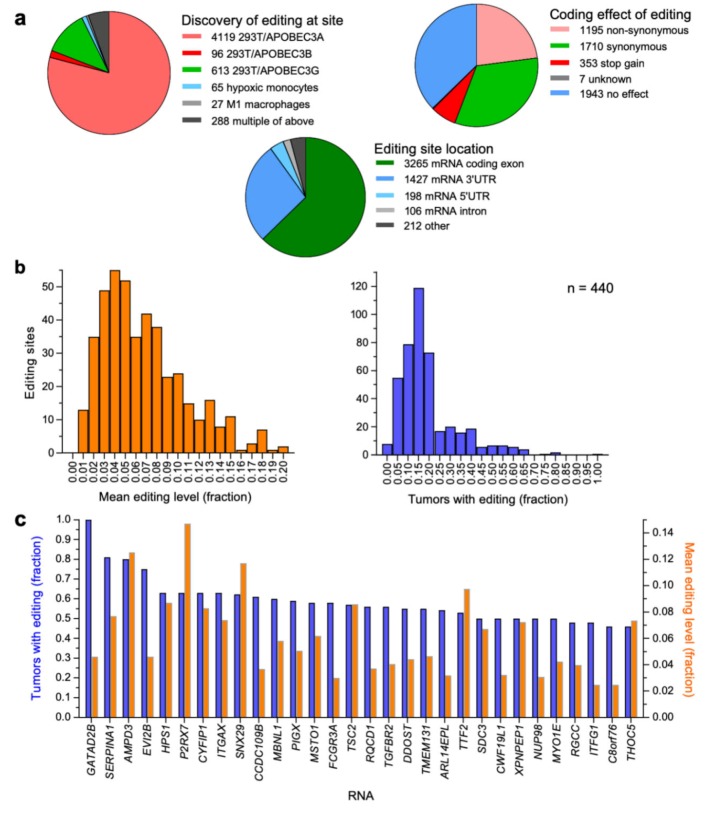
APOBEC3-mediated RNA editing sites. (**a**) Characteristics of the sites that are examined in this study are illustrated with pie-charts. UTR, untranslated region. (**b**) Frequency histograms are shown for the fraction of The Cancer Genome Atlas (TCGA)-BRCA tumors with detectable RNA editing and the mean level of such editing for 440 sites for which editing was detected in the tumor cohort. (**c**) Fraction of tumors with editing and their mean editing level are shown for the 30 most commonly edited RNAs of the cohort.

**Figure 2 ijms-20-05621-f002:**
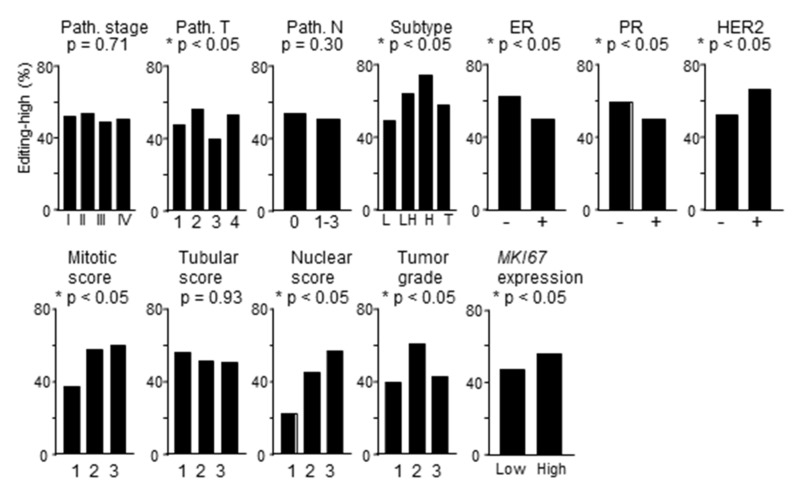
RNA editing and clinico-pathological features. Incidence of RNA editing-high tumors are plotted for different features of breast cancer. Pathological (Path.) stage and T and N TNM values are as per the American Joint Committee on Cancer staging system. Indicators of breast cancer subtypes are as follows: L, luminal type; LH, luminal-HER2; H, HER2; T, triple negative. For *MKI67* gene expression, the median gene expression value was used to identify low and high groups. *p*-values were determined with Fisher′s exact test. ER, estrogen receptor; PR, progesterone receptor; HER2, human epidermal growth factor receptor 2.

**Figure 3 ijms-20-05621-f003:**
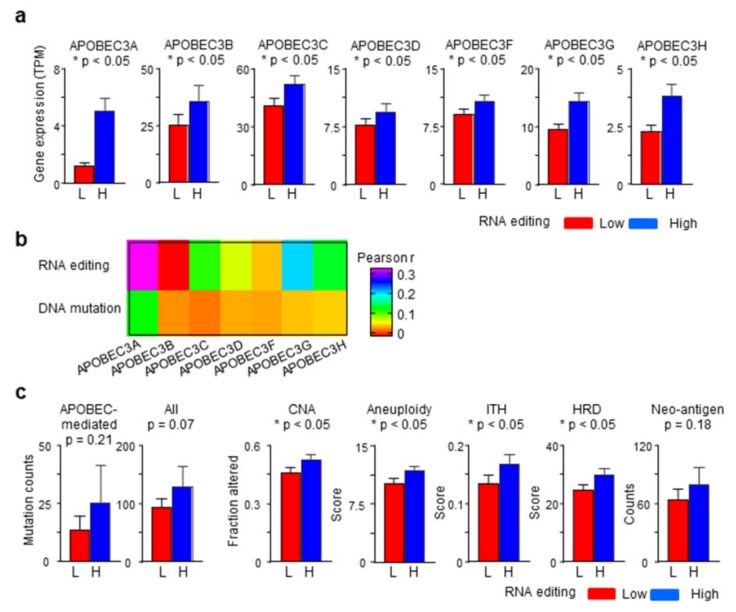
RNA editing and APOBEC3 gene expression and genome characteristics of tumors. (**a**) Tukey boxplots show *APOBEC3* gene expression of RNA editing-high (H) and -low (L) tumors. TPM, transcripts per million. (**b**) Heatmap shows Pearson′s correlation coefficients for association of *APOBEC3* gene expression levels with APOBEC3-mediated C-to-U RNA editing score and APOBEC signature DNA mutation burden of tumors. (**c**) Barplots are shown for comparison of tumor genome characteristics of the RNA editing-high and -low tumors. *p*-values in two-group comparisons were calculated with Fisher’s exact or standard *t*-tests. CNA, copy number alteration; HRD, homologous recombination deficiency; ITH, intra-tumor heterogeneity; SNV, single nucleotide variation.

**Figure 4 ijms-20-05621-f004:**
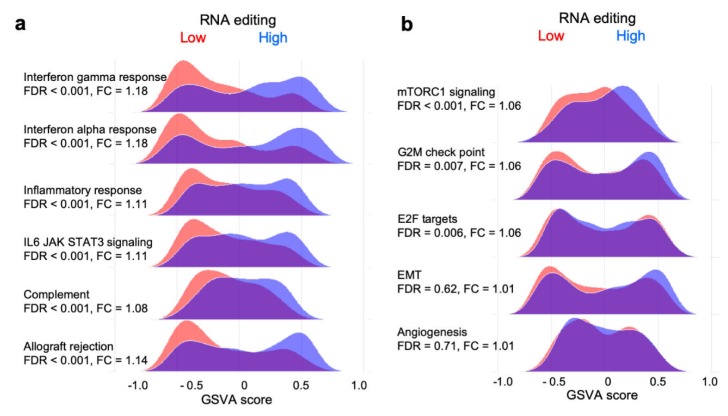
Immune-related gene sets have enriched expression in RNA editing-high tumors. Shown are examples of ridgeline plots of gene set variation analysis (GSVA) scores of RNA editing-high and -low tumors for (**a**) six immune-related gene sets and (**b**) five other gene sets of Molecular Signatures Database (MSigDB) Hallmark collection. Fold change (FC) values comparing scores of editing-high and -low tumors and false discovery rate (FDR) values in significance analysis of the comparisons were calculated with a Bayes-moderated *t*-test. EMT, epithelial mesenchymal transition.

**Figure 5 ijms-20-05621-f005:**
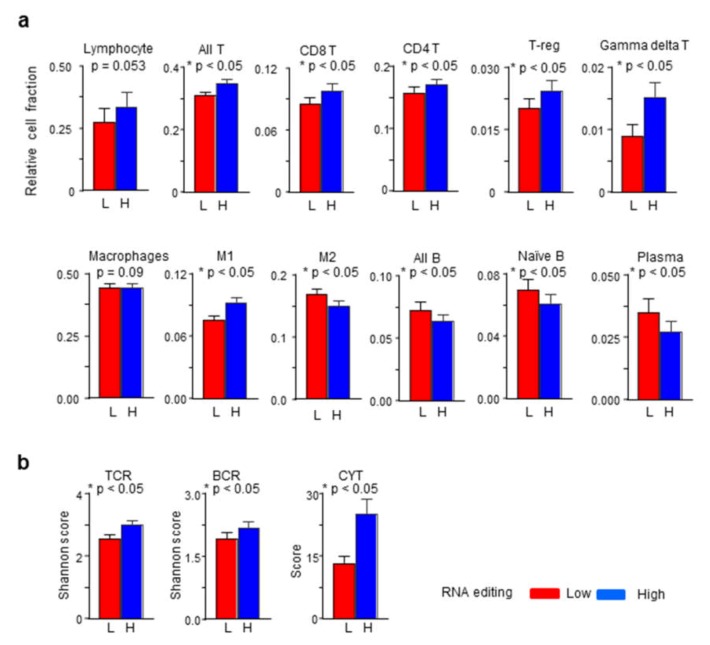
RNA editing and immune microenvironment features. Barplots of values for (**a**) relative fractions of tumor-infiltrating immune cells of different types and (**b**) T cell receptor (TCR) and B cell receptor (BCR) diversity (Shannon scores) and immune cytolytic activity (CYT score) are shown for the groups of RNA editing-high (H) and -low (L) tumors. The *p*-values in two-group comparisons were calculated with standard *t*-test.

**Figure 6 ijms-20-05621-f006:**
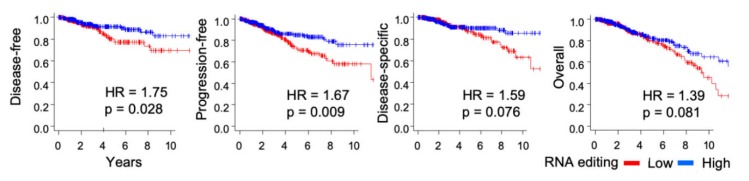
RNA editing is associated with improved survival. Kaplan–Meier survival plots along with log-rank test *p*-value and Cox proportional hazard ratio (HR) value are shown for association of RNA editing in tumors with disease-free, progression-free, disease-specific, and overall survival of patients. Groups of RNA editing-high and -low cases are compared.

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
