# Peer review of "APOBEC3-Mediated RNA Editing in Breast Cancer is Associated with Heightened Immune Activity and Improved Survival"

_ijms, 2019, doi:10.3390/ijms20225621_

Round 1

Reviewer 1 Report

Review for Manuscript ijms-633615-peer-review-v1

General Comments: Extremely well designed, written, and presented, which made the review process extremely easy. A few general comments:

1) There is a lot of methods in the results, which are then duplicated in what they say in the methods section. This makes for a very long results section which could be shortened by removing some of this duplication.

2) While gene expression changes are suggestive of differential changes in certain immune cell populations, the analysis is performed on samples of solid tumors, which have a mixed immune population and these are at varying stages of differentiation/maturation. Discuss that this is a limitation in making conclusions about specific cell types when compared to flow cytometric or histologic identification and quantification.

More specific comments are listed below by line number.

More Specific Comments:

Title – None

Abstract – None

Introduction – None

Results:

1) Line 163 – “American Joint Committee on Cancer” is in a different sized font than the other text.

Discussion – See general comments above.

Methods – See comments above about Results section containing Methods.

Figures, Tables, and Legends:

1) Check the font alignment underneath the bars in the subfigures for Figure 2. Many are not centered under the bars.

Author Response

Reviewer 1

General Comments: Extremely well designed, written, and presented, which made the review process extremely easy. A few general comments:

1) There is a lot of methods in the results, which are then duplicated in what they say in the methods section. This makes for a very long results section which could be shortened by removing some of this duplication.

>> Thank you for your kind comments. We had now modified and delated the duplicated sentences in the results.

2) While gene expression changes are suggestive of differential changes in certain immune cell populations, the analysis is performed on samples of solid tumors, which have a mixed immune population and these are at varying stages of differentiation/maturation. Discuss that this is a limitation in making conclusions about specific cell types when compared to flow cytometric or histologic identification and quantification.

>> We thank the reviewer for this valuable suggestion. We had now added the limitations for the method which we performed to estimate the immune cell populations in tumor from RNA sequenced data.

Discussion part (Line 341-348) ------“In this study, we used one of the established methods “CIBERSORT” to estimate tumor-infiltrated immune cells using RNA sequenced data from bulk tumor which containing noise or unknown cells, also closely related call types. Limitations for this algorism is the accuracy of parameters of cell profiles, which could be affected by undergoing heterotypic associations, phenotypic malleability, or disease-induced deficiency of regulations of cell differentiation. Therefor some specific cell types could be over- or under-estimated by this method, and the differences of cell population in tumors compared to flow cytometric or histologic identification and quantification would be occurred.”

More specific comments are listed below by line number.

More Specific Comments:

Results:

1)         Line 163 – “American Joint Committee on Cancer” is in a different sized font than the other text.

>> We had now revised the word font and size in the results section.

Discussion – See general comments above.

Methods – See comments above about Results section containing Methods.

Figures, Tables, and Legends:

1)         Check the font alignment underneath the bars in the subfigures for Figure 2. Many are not centered under the bars.

>> We had now aligned underneath the bars in the Figure 2.

Reviewer 2 Report

Authors improved the quality of the manuscript addressing my suggestions.

Author Response

Reviewer 2

Authors improved the quality of the manuscript addressing my suggestions.

>> We are extremely thankful for your kind comments.Thank you very much.

Round 2

Reviewer 1 Report

The authors have nicely addressed all comments.

For the new passage in the Discussion: Therefor some specific cell types could be over- or underestimated by this method, and the differences of cell population in tumors compared to flow cytometric or histologic identification and quantification would be occurred.

1) Change "Therefor" to "Therefore"

2) The passage "and the differences" to the end of this sentence needs to be reworded since it is currently unclear in its meaning.

Author Response

We thank reviewer #1 suggesting new changes to the manuscript. All the changes were suggested for one paragraph in Discussion. We have now incorporated the two changes:

Spelling error was corrected for 'therefore.' The last sentence was edited to make it meaningful.

We have also made a few other minor changes to the paragraph to improve its readability. The updated paragraphs is as follows:

In this study, we used a well-established method, CIBERSORT, to estimate tumor-infiltrating immune cells using RNA sequencing data of whole tumor, which, besides cancer cells, contains unknown cells as well as closely related types of cells such as immune cell subsets. Accurate estimation of tumor immune cells with this algorithm has limitations because of inadequacies in cell type signatures that the algorithm relies on, phenotypic malleability, and disease-induced alterations in cellular differentiation or transcription. Therefore, abundances of some specific tumor-infiltrating immune cell types could have been over- or under-estimated in our study, and the associations of abundances with RNA editing that we observed may be different if infiltrating immune cells had been quantified by flow cytometric or histologic methods.

This manuscript is a resubmission of an earlier submission. The following is a list of the peer review reports and author responses from that submission.

Round 1

Reviewer 1 Report

Review for Manuscript ijms-605602-peer-review-v1

General Comments: Extremely well designed, written, and presented, which made the review process extremely easy. A few general comments:

In the abstract and body, looking at line 27 of the abstract, do the percentages and ranges of values reflect the 411 genes or 440 sites? Please clarify throughout the manuscript. When discussing low or high cutoffs for editing, clarify the cutoff values throughout the manuscript. There is a lot of methods in the results, which are then duplicated in what they say in the methods section. This makes for a very long results section which could be shortened by removing some of this duplication. While gene expression changes are suggestive of differential changes in certain immune cell populations, the analysis is performed on samples of solid tumors, which have a mixed immune population and these are at varying stages of differentiation/maturation. Discuss that this is a limitation in making conclusions about specific cell types when compared to flow cytometric or histologic identification and quantification. Check for side by side double spaces between words rather than single spaces between words. There are a few in the manuscript.

More specific comments are listed below by line number.

More Specific Comments:

Title – None

Abstract – None

Introduction – None

Results

For “similar editing scores as suggested by a P value of 0.06”, is this implying that there is a similarity since the P value is non-significant at 0.06? Explain further the importance of this P value since it is almost at the significance threshold of 0.05. For the triple negative subtype, there is a lower percentage of “editing high”. Since triple negative tumors historically have been identified as having a worse prognosis, expand on this.

Discussion – See general comments above.

Methods

Line 437 – Remove “were” after “values”

Figures, Tables, and Legends:

Figures 1 and 3 are missing from the PDF. Check the font alignment underneath the bars in the subfigures for Figure 2. Many are not centered under the bars.

Reviewer 2 Report

Comments

Language style

The current manuscript is not well- written. In general, substandard writing and numerous grammatical errors make it difficult to understand how the authors have interpreted the cutting-edge discoveries.

Statistical assay issue

2.1 Conceptual errors

According to figure legends for Fig.3E and Fig.5, related panels should be turkey box plot, however, the real images are bar graph type. Regarding to individual cohort, the exact samples number should be highlighted.

2.2 Image editing

Regarding to description of p value, whether authors could use typical makers to label, such as “* p < 0.05”, etc.

Reviewer 3 Report

In the present form, it is difficult to appreciate the quality of the manuscript.

Authors referred to BRCA cancers but in the introduction section no specification about estrogen, progesterone and /or EGF receptor, and also androgen receptor that displays a relevant action in triple negative breast cancers is described (see DOI: 10.3389/fendo.2018.00492). Authors describe the results in a detailed manner that is not direct to a broad audience. They should summarize paragraph 1 and 2 and focus their attention on a subset of genes or a gene whose sequence is edited and on their impact on breast cancer onset and progression. If the editing rate increases is not appealing. Instead is appealing know the gene edited in cancer. As a consequence, authors should emphasize figure 2. Again, data about immune activity seems to be not persuasive. What is the impact of editing on gene expression and/or protein function?

Minor concern:

HEK-293T cell are not fibroblasts